# Asymptomatic Malaria Infections in the Time of COVID-19 Pandemic: Experience from the Central African Republic

**DOI:** 10.3390/ijerph19063544

**Published:** 2022-03-16

**Authors:** Emilia Bylicka-Szczepanowska, Krzysztof Korzeniewski

**Affiliations:** 14th Department of Infectious Diseases, Provincial Hospital for Infectious Diseases, 01-201 Warsaw, Poland; emilia.bylicka@wp.pl; 2Department of Epidemiology and Tropical Medicine, Military Institute of Medicine, 04-141 Warsaw, Poland; 3Department of Tropical Medicine and Epidemiology, Institute of Maritime and Tropical Medicine, Medical University of Gdańsk, 81-519 Gdynia, Poland

**Keywords:** asymptomatic malaria, mRDT, Central African Republic, COVID-19 pandemic

## Abstract

According to the latest World Health Organization malaria report, 95% of 241 million global malaria cases and 96% of 627,000 malaria deaths that were recorded in 2020 occurred in Africa. Compared to 2019, 14 million more cases and 69,000 more malaria deaths were recorded, mainly because of disruptions to medical services during the COVID-19 pandemic. The aim of this study was to assess the prevalence of asymptomatic malaria cases in children and adults living in the Dzanga Sangha region in the Central African Republic (CAR) during the COVID-19 pandemic. Rapid immunochromatographic assays for the qualitative detection of *Plasmodium* species (*P. falciparum*, *P. vivax*, *P. ovale*/*P. malariae*) circulating in whole blood samples were used. A screening was performed in the group of 515 patients, 162 seemingly healthy children (aged 1–15) and 353 adults, all inhabiting the villages in the Dzanga Sangha region (southwest CAR) between August and September 2021. As much as 51.2% of asymptomatic children and 12.2% of adults had a positive result in malaria rapid diagnostic tests (mRDTs). Our findings demonstrated a very high prevalence of asymptomatic malaria infections in the child population. Limited access to diagnostics, treatment and prevention of malaria during the global COVID-19 pandemic and less medical assistance from developed countries may be one of the factors contributing to the increase in the prevalence of disease in Africa.

## 1. Introduction

Malaria is a vector-borne parasitic disease, which in humans, is primarily caused by one of the five species of *Plasmodium*: *Plasmodium falciparum*, *Plasmodium vivax*, *P. ovale (P. ovale curtisi*, *P. ovale wallikeri)*, *P. malariae* and *P. knowlesi* (simian species found in Southeast Asia). Thanks to recent developments in molecular diagnostics, it is possible to identify further enzootic species of *Plasmodium* in humans: *P. simium*, *P. cynomolgi*, *P. coatneyi* and *P. inui*; however, their prevalence and clinical picture have not been conclusively established [1,2].

According to the latest World Health Organization (WHO) report on malaria (The World Malaria Report 2021) [3], there were an estimated 241 million malaria cases in 2020 in 85 endemic countries, increasing from 227 million in 2019, with 95% of the cases reported from Africa (>50% of global malaria infections are reported from only five countries: Nigeria, the Democratic Republic of the Congo, Uganda, Mozambique and Angola). The number of malaria deaths globally in 2020 was 627,000 (69,000 deaths more compared to 2019), mainly because of disruptions in the provision of medical services during the coronavirus disease 2019 (COVID-19) pandemic. It is worth pointing out that 96% of global malaria deaths are reported in Africa (with Nigeria, the Democratic Republic of the Congo, Uganda, Mozambique and Angola accounting for >50% of deaths). Most malaria infections and deaths on the African continent occur in children under 5 years old [3].

Some experts estimate that because of the effects of the ongoing COVID-19 pandemic and a possible collapse of the public health care systems in many developing countries, the rates of morbidity and mortality of malaria in Africa are likely to increase rapidly [4]. According to WHO, *P. falciparum* is responsible for 99.7% of all malaria infections in Africa [3]. In 2020 there were a total of 215 million suspected, presumed or confirmed cases of malaria on the continent [3]. However, some experts estimate that a lot of malaria cases in Africa go undetected (this is associated with a high proportion of asymptomatic infections), and this, in turn, remains a major obstacle in the fight to eliminate malaria from the continent [4]. The Central African Republic (CAR) is a landlocked country in Sub-Saharan Africa where malaria transmission occurs year-round. The population of the country is 4.8 million people. According to WHO, there were 2,730,158 suspected and 1,980,804 presumed or confirmed cases of malaria in CAR in 2020; all of the cases were reportedly caused by *P. falciparum* [3]. In CAR, malaria is considered to be responsible for as much as 40% of all reported illnesses and 10% of all registered deaths, especially in the paediatric population [5]. Because diagnostic capabilities in CAR are limited and because of high rates of asymptomatic infections in the area, a significant proportion of cases are classified as suspected. In early 2020 (in the pre-pandemic period), the authors tested for malaria 500 seemingly healthy children (aged 1–15) living in the Dzanga Sangha region in southwest CAR (the same population of children was involved in the present study). Rapid diagnostic test (RDT) results showed that the overall prevalence of asymptomatic malaria in the study group at that time was 35.2% [6]. 

The present study aimed to assess the prevalence of asymptomatic malaria cases in children and adults living in the Dzanga Sangha region in the Central African Republic during the COVID-19 pandemic.

## 2. Materials and Methods

### 2.1. Study Population 

The study was conducted between August and September 2021 and involved a group of 515 patients, children (aged 1–15 years) and adults (aged 16–76) of both sexes (settled Bantu and semi-nomadic BaAka Pygmies) living in the Monasao village (population of approx. 4000 people) and neighbouring villages in the Dzanga Sangha region (Bayanga subprefecture with a population of approx. 12,000; Sangha-Mbaere prefecture; elevation: 510 m above sea level) in the southwest CAR. Adults and children were recruited in the villages, then directed to the local health care facility in Monasao, where malaria rapid diagnostic tests (mRDTs) were used (the only available diagnostic tool for malaria detection, no possibility of malaria confirmation by microscopy). The only inclusion criteria were the absence of clinical signs and symptoms of malaria (body temperature ≤37.5 °C). The exclusion criteria were: the presence of malaria signs and symptoms, anti-malarial treatment received in the past 28 days and difficulties in the venepuncture procedure. On behalf of the paediatric patients, the consent to participate in the study was given by their parents. The parents had been informed of the study’s purpose and methods. The interviews were conducted in their native language, either in Sango (a language spoken by the Bantu) or Mbenzele (a language used by the BaAka Pygmies). Medical personnel responsible for carrying out the study procedures measured the participants’ body weight and body temperature (correct temperature was defined as ≤37.0 °C, low-grade fever: 37.1–37.5 °C, fever: ≥37.6 °C). Next, the medical staff performed RDTs for malaria as well as haemoglobin measurements to identify any possible cases of anaemia.

### 2.2. Malaria Screening and Blood Sampling Procedures

Immunochromatographic mRDT (Pf/Pv/Pan; Beright, Hangzhou AllTest Biotech Co., Ltd.) detecting *Plasmodium* species (*P. falciparum*, *P. vivax*, *P. ovale/P. malariae*) in whole blood samples were used. Asymptomatic malaria was defined as the presence of *Plasmodium* species on mRDT with no clinical signs of the disease (body temperature ≤37.5 °C, no chills, headache, joint pain, weakness, vomiting or diarrhoea). The mRDT sensitivity (the ability of the test to correctly identify patients who have malaria, true positive rate) is >98.7% and its specificity (the ability of the test to correctly identify patients who do not have malaria, true negative rate) is >99.0% (according to the manufacturer’s specifications) [7].

### 2.3. Haemoglobin Measurements

Haemoglobin level was measured using a portable DiaSpect^TM^ analyzer (EKF Diagnostics, Cardiff, UK) by examining a venous blood sample. The definition of anaemia was based on the WHO criteria [8], where a haemoglobin level of ≥12 g/dL in women, ≥13 g/dL in men and ≥12 g/dL in elderly people (both sexes) is regarded as normal; 11.0–11.9 g/dL in women and 11.0–12.9 g/dL in men as mild, 8.0–10.9 g/dL as moderate and <8.0 g/dL as severe anaemia.

### 2.4. Statistical Methods

Statistical analysis was carried out using StatSoft, Inc. (2014) STATISTICA (data analysis software system) version 12.0. (StatSoft, Kraków, Polska) www.statsoft.com (accessed on 10 October 2021). All data were presented as mean (standard deviation), range, median and 95% confidence interval (CI). Groups of data were compared using t-Student, U Mann-Whitney or Kruskal-Wallis test. Values of *p* < 0.05 or lower were considered significant.

### 2.5. Ethical Approval

The research project was approved by the Committee on Bioethics at the Military Institute of Medicine, Warsaw, Poland (Decision No. 22/WIM/2020) under the Declaration of Helsinki (1996) and the rules elaborated by the European Union “Good clinical practice for trials on medicinal products in the European Community. The rules governing medicinal products in the European Community” (1999) ratified by the Committee of Ethics in Poland (March 1993). The tests performed in the CAR were carried out with the written consent of each participant and under the supervision of priest Wojciech Lula (a catholic mission superior and a manager of the healthcare centre in Monasao, Bayanga subprefecture, Sangha-Mbaere prefecture), with considerable help from the medical staff working at the healthcare centre.

## 3. Results

The study involved 162 children (Table 1) and 353 adults (Table 2) without any clinical signs of malaria. The most significant finding of the study was a high prevalence of malaria (51.2%) in asymptomatic children. The mean age of the children with a positive result on mRDT was 5.8 years. Children with mRDT (+) were statistically significantly younger (*p* = 0.0006). The mean body weight of the children with mRDT (+) was 17.0 kg. Children with mRDT (+) had significantly lower body weight (*p* < 0.0001). The mean body temperature of the children with mRDT (+) as well as those with mRDT (–) was 36.6 °C. The mean Hb g% of the children with mRDT (+) was 10.4 g/dL. Children with mRDT (+) had significantly lower Hb g% (*p* < 0.000001).

The prevalence of *Plasmodium* infections was much lower among asymptomatic adults than it was among children (12.2% vs. 51.2%, *p* < 0.0001). The mean Hb g% of the adults with mRDT (+) was 11.5. Adults with mRDT (+) had significantly lower Hb g% (*p* = 0.0036).

The study revealed infections with different species of *Plasmodium*, including *P. falciparum* (99.0%), *P. vivax* (6.3%) and Pan (*P. ovale*/*P. malariae*, 4.8%) (Table 3 and Table 4). More than one *Plasmodium* species were observed in 3.6% of the children and 11.6% of the adults tested.

## 4. Discussion

Malaria remains a primary health issue in Africa as it accounts for 9% of all illnesses reported from this continent [3]. In Europe, more than 8000 cases of imported malaria are reported each year, and a vast majority of those cases are imported from Africa [9]. The WHO has warned that due to the COVID-19 impact, the number of new cases and deaths from malaria in Africa is likely to increase. On the other hand, because the prevalence rates of SARS-CoV-2 infections are much lower in African countries than they are in Asia, Europe or in North and South America [10], tourism industry experts forecast that Africa may soon become the popular destination for travellers from high-income countries (12% increase in arrivals in 2021 compared to 2020) [11]. 

The aim of this study was to assess the prevalence of asymptomatic malaria cases in children and adults living in the Dzanga Sangha region in the Central African Republic (CAR) during the COVID-19 pandemic. The WHO has indicated that the actual rates of malaria cases in the CAR could be underestimated, and in fact, they may be several times higher than the official reports suggest. This is mainly due to the fact that a significant number of malaria cases in the region are asymptomatic [12]. Children are one of the most vulnerable groups affected by malaria in high-transmission areas of sub-Saharan Africa [13]. Asymptomatic carriers do not seek treatment for their infection and add up to a reservoir of parasites available for transmission by *Anopheles* mosquitoes [14]. The long-term asymptomatic carriage may represent a form of tolerance to the parasite in children building up their immune response. In this way, the asymptomatic carriage would protect these children from developing a mild or severe malaria attack by keeping their immunity effective [15,16]. Conversely, asymptomatic carriage may represent a mode of entry to symptomatic malaria, especially in young children [17].

The results of this study conducted during the COVID-19 pandemic (August–September 2021) revealed that as much as 51.2% of the involved paediatric group (i.e., 162 seemingly healthy children aged 1–15 years old living in the southwest Central African Republic) had asymptomatic malaria. A study conducted before the pandemic (early 2020) in the same group of children (500 children aged 1–15) identified asymptomatic cases of malaria in 35.2% of the examined group. The results demonstrate very high malaria prevalence rates in the child population living in Central Africa. Our current findings are similar to those reported by Maziarz et al. [18] in northern Uganda in a sample of more than 1000 children under 16 years old, where asymptomatic malaria was identified in 52.4% participants. Asymptomatic infections go unnoticed and thus are never treated [19], resulting in anaemia, disturbed concentration and learning difficulties [20]. They are prevalent in all parts of sub-Saharan Africa [21,22,23] and pose a real challenge for malaria prevention and control strategies on the continent. Asymptomatic carriers remain the main reservoir of infection that spawns new clinical cases. The mean age of asymptomatic children with a positive result on mRDT in our research study was 5.8 years. Children with mRDT (+) were statistically significantly younger. According to Ndamukong-Nyanga et al. [24], the drop in the prevalence of malaria associated with an increase in age is related to the acquisition of protective immunity due to repeated infections as children grow older in high transmission areas. These findings have been confirmed by the results of a study conducted in Tanzania which showed a reduction in positive mRDTs among older children [25]. The mean Hb % in asymptomatic children with mRDT (+) was found to be 10.4 g/dL (range 6.5–13.8) vs. 11.6 (range 8.8–15.5) in children with mRDT (–). Our study has confirmed a high prevalence of mild to moderate anaemia in the paediatric population involved in the study. Children with mRDT (+) had significantly lower Hb %. These findings are comparable with the results obtained by Teh et al. [26], who found that there were more positive mRDTs among anaemic than non-anaemic children. Frequent anaemia has also been confirmed by Nzobo et al., who aimed to assess the prevalence of malaria in a group of >300 asymptomatic Tanzanian school-children aged 6–13 years and found that their mean haemoglobin level was 10.1 g/dL [27].

The present study revealed that the prevalence of *Plasmodium* infections was much lower among asymptomatic adults (n = 353) than it was among children. The study found that only 12.2% of adults were infected with malaria. A lower prevalence rate in this group could be associated with protective antiparasitic immunity. In areas of high malaria transmission, the frequency of clinical events and average parasite densities are likely to decrease with age in line with increasing acquired immunity [28]. Subsequent *Plasmodium* infections are likely to become less severe and manifest with milder signs and symptoms. They may also be asymptomatic or take the form of infection with low subpatent parasite densities below the limit of detection (50–100 parasites/µL) by mRDT, depending on the gradual development of protective immunity and different degree of antigenic diversity. Age seems to be an important factor affecting the longevity of the humoral response, as adults seem to make better long-lived antibody responses to *Plasmodium* antigens (antibodies protect from reinfection) than children [29]. 

Disruption in the provision of diagnosis, prevention and treatment remains a major obstacle in the fight to eliminate malaria and other tropical diseases from Africa. The same may be true for the efforts undertaken to combat the COVID-19 pandemic. The WHO vaccination strategy assumed that a minimum of 40% of each country’s population would receive full primary immunisation against SARS-CoV-2 by the end of 2021 [30]. To date, only 9% of Africa’s population has been vaccinated against COVID-19 [30]. According to the WHO, Africa is the least affected region globally in terms of the number of reported SARS-CoV-2 infections (1.5%) and the number of COVID-19 deaths (0.1%) [31]. COVID-19 is the sixth leading cause of death in high-income countries and the 41st cause of death in sub-Saharan Africa [32]. Africa’s low morbidity and mortality rates are primarily due to a lower population median age and a lower obesity prevalence compared to high-income countries in Europe or North America [31]. On the other hand, a small number of confirmed cases may be attributable to limited testing. Focusing on the COVID-19 pandemic and diverting resources from treating other conditions may slow down the fight against infectious diseases which are endemic in Africa [31].

Limitations of the study: no information on cases of malaria among pregnant women in the study group, lack of microscopy data and PCR confirmation of malaria infections (low sensitivity of RDT compared to PCR may lead to false-negative results and an underestimation of the prevalence rate of asymptomatic *Plasmodium* carriers), possible long-term positivity of HRP2-based mRDTs even after complete elimination of *P. falciparum* from patients with effective treatment.

## 5. Conclusions

The current prevalence of asymptomatic malaria in children living in the southwest of the Central African Republic is very high. In contrast, the prevalence of asymptomatic malaria in adults studied at the same time and place was found to be several times lower than in the paediatric population, which might be attributable to their acquired protective immunity against *Plasmodium* infections. The WHO has warned that there may be a further increase in malaria-related morbidity and mortality in Africa in the following years, as a consequence of diverting funding from control strategies aimed at diagnosis, prevention and treatment of tropical diseases (including malaria) to the COVID-19 control strategies.

## Figures and Tables

**Table 1 ijerph-19-03544-t001:** Characteristics of asymptomatic children (*n* = 162).

Variables	mRDT (−) (*n* = 79)	mRDT (+) (*n* = 83)	Total (*n* = 162)	*p*-Value
Age (years)				0.0006 ^1^
Mean (SD)	7.8 (3.9)	5.8 (3.6)	6.8 (3.9)	
Range	1.0–15.0	1.0–15.0	1.0–15.0	
Median	8.0	5.0	6.0	
95%CI	[7.0; 8.7]	[5.0; 6.6]	[6.2; 7.4]	
Sex				0.3301 ^3^
Female	46 (58.2%)	42 (50.6%)	88 (54.3%)	
Male	33 (41.8%)	41 (49.4%)	74 (45.7%)	
Body weight (kg)				<0.0001 ^1^
Mean (SD)	22.8 (10.2)	17.0 (9.2)	19.8 (10.1)	
Range	4.0–49.0	3.0–50.0	3.0–50.0	
Median	21.0	15.0	18.0	
95%CI	[20.5; 25.1]	[15.0; 19.0]	[18.2; 21.4]	
Body temperature (°C)				0.6534 ^1^
Mean (SD)	36.6 (0.3)	36.6 (0.3)	36.6 (0.3)	
Range	36.0–37.4	36.0–37.4	36.0–37.4	
Median	36.7	36.7	36.7	
95%CI	[36.6; 36.7]	[36.6; 36.7]	[36.6; 36.7]	
Hb g%				<0.000001 ^2^
Mean (SD)	11.6 (1.4)	10.4 (1.5)	11.0 (1.6)	
Range	8.8–15.5	6.5–13.8	6.5–15.5	
Median	11.6	10.4	11.1	
95%CI	[11.3; 12.0]	[10.1; 10.8]	[10.8; 11.3]	

^1^ U Mann-Whitney; ^2^
*t*-Student; ^3^ Chi-square.

**Table 2 ijerph-19-03544-t002:** Characteristics of asymptomatic adults (*n* = 353).

Variables	mRDT (−) (*n* = 310)	mRDT (+) (*n* = 43)	Total (*n* = 353)	*p*-Value
Age (years)				0.2004 ^1^
Mean (SD)	37.8 (14.8)	35.5 (16.9)	37.5 (15.1)	
Range	16.0–76.0	16.0–76.0	16.0–76.0	
Median	37.5	32.0	36.0	
95%CI	[36.1; 39.4]	[30.3; 40.7]	[35.9; 39.1]	
Sex				0.2288 ^2^
Female	202 (65.2%)	32 (74.4%)	234 (66.3%)	
Male	108 (34.8%)	11 (25.6%)	119 (33.7%)	
Body weight (kg)				0.8601 ^1^
Mean (SD)	47.9 (10.1)	47.4 (8.4)	47.8 (9.9)	
Range	22.0–105.0	32.0–69.0	22.0–105.0	
Median	46.0	46.0	46.0	
95%CI	[46.8; 49.0]	[44.9; 50.0]	[46.8; 48.9]	
Body temperature (°C)				0.4034 ^1^
Mean (SD)	36.6 (0.2)	36.5 (0.3)	36.6 (0.2)	
Range	36.0–37.4	36.0–37.1	36.0–37.4	
Median	36.6	36.6	36.6	
95%CI	[36.6; 36.6]	[36.5; 36.6]	[36.6; 36.6]	
Hb g%				0.0036 ^1^
Mean (SD)	12.2 (1.7)	11.5 (1.4)	12.1 (1.7)	
Range	9.3–16.5	9.0–14.5	9.0–16.5	
Median	12.2	11.5	12.2	
95%CI	[12.0; 12.4]	[11.0; 11.9]	[11.9; 12.3]	

^1^ U Mann-Whitney; ^2^ Chi-square.

**Table 3 ijerph-19-03544-t003:** Children with mRDT (+) (*n* = 83).

Variables	*P. falciparum* (*n* = 83)	*P. vivax* (*n* = 3)	Pan (*P. ovale*/*P. malariae*) (*n* = 1)
Age (years)			
Mean (SD)	5.8 (3.6)	7.7 (4.6)	13 years
Range	1.0–15.0	5.0–13.0	
Median	5.0	5.0	
95%CI	[5.0; 6.6]	[−3.8;19.1]	
Sex			
Female	42 (50.6%)	2 (66.7%)	1 (100.0%)
Male	41 (49.4%)	1 (33.3%)	0 (0.0%)
Body weight (kg)			
Mean (SD)	17.0 (9.2)	20.0 (8.7)	30 kg
Range	3.0–50.0	14.0–30.0	
Median	15.0	16.0	
95%CI	[15.0; 19.0]	[−1.7;41.7]	
Body temperature (°C)			
Mean (SD)	36.6 (0.3)	36.4 (0.1)	36.5 °C
Range	36.0–37.4	36.4–36.5	
Median	36.7	36.4	
95%CI	[36.6; 36.7]	[36.3;36.6]	
Hb g%			
Mean (SD)	10.4 (1.5)	10.9 (1.4)	9.3 g%
Range	6.5–13.8	9.3–11.7	
Median	10.4	11.7	
95%CI	[10.1; 10.8]	[7.5;14.3]	

**Table 4 ijerph-19-03544-t004:** Adults with mRDT (+) (*n* = 43).

Variables	*P. falciparum* (*n* = 43)	*P. vivax* (*n* = 5)	Pan (*P. ovale*/*P. malariae*) (*n* = 5)	*p*-Value
Age (years)				0.1478 ^1^
Mean (SD)	35.5 (16.9)	46.6 (17.6)	46.6 (17.6)	
Range	16.0–76.0	22.0–65.0	22.0–65.0	
Median	32.0	48.0	48.0	
95%CI	[30.3; 40.7]	[24.7;68.5]	[24.7;68.5]	
Sex				0.9733 ^2^
Female	32 (74.4%)	4 (80.0%)	4 (80.0%)	
Male	11 (25.6%)	1 (20.0%)	1 (20.0%)	
Body weight (kg)				0.1664 ^1^
Mean (SD)	47.4 (8.4)	52.0 (8.5)	52.0 (8.5)	
Range	32.0–69.0	43.0–66.0	43.0–66.0	
Median	46.0	50.0	50.0	
95%CI	[44.9; 50.0]	[41.5;62.5]	[41.5;62.5]	
Body temperature (°C)				0.3705 ^1^
Mean (SD)	36.5 (0.3)	36.7 (0.3)	36.7 (0.3)	
Range	36.0–37.1	36.5–37.1	36.5–37.1	
Median	36.6	36.6	36.6	
95%CI	[36.5; 36.6]	[36.4;37.1]	[36.4;37.1]	
Hb g%				0.9751 ^1^
Mean (SD)	11.5 (1.4)	11.8 (0.4)	11.8 (0.4)	
Range	9.0–14.5	11.3–12.2	11.3–12.2	
Median	11.5	11.9	11.9	
95%CI	[11.0; 11.9]	[11.3;12.2]	[11.3;12.2]	

^1^ Kruskal-Wallis; ^2^ Chi-square.

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
