# Peer review of "Asymptomatic Malaria Infections in the Time of COVID-19 Pandemic: Experience from the Central African Republic"

_ijerph, 2022, doi:10.3390/ijerph19063544_

Round 1

Reviewer 1 Report

The manuscript by Bylicka-Szczepanowska and Korzeniewski presents interesting analysis regarding the asymptomatic malaria infections in the time of COVID-19 2 pandemic. The paper is appropriate for Int. J. Environ. Res. Public Health. However, the manuscript needs to be revised before publication (Note line numbers correspond to the small numbers on the side of the page):

  1. Page 1, Line 5: ”4th Department of Infectious Diseases, …” make sure this is correct.
  2. P2 L69: “RDT” to “rapid diagnostic tests (RDT)”
  3. P2 L82: “mRDT” to “malaria rapid diagnostic tests (mRDTs)”
  4. P2 L81: “Central African Republic” to “CAR”
  5. P3 L97: “Immunochromatographic malaria rapid diagnostic tests (mRDT Pf/Pv/Pan; Beright,” to  “Immunochromatographic mRDT (Pf/Pv/Pan; Beright,”  
  6. P3 L124: “Central African Republic” to “CAR”; also change others

Author Response

Dear Reviewer,

Thank you very much for your review and valuable comments.
I present the corrections made below:

 1. Page 1, Line 5: ”4th Department of Infectious Diseases, …” make sure this is correct.
The first author's affiliation is correct and has been used in the publication of previous manuscripts

2. P2 L69: “RDT” to “rapid diagnostic tests (RDT)”
Done

3. P2 L82: “mRDT” to “malaria rapid diagnostic tests (mRDTs)”
Done

4. P2 L81: “Central African Republic” to “CAR”
Done

5. P3 L97: “Immunochromatographic malaria rapid diagnostic tests (mRDT Pf/Pv/Pan; Beright,” to  “Immunochromatographic mRDT (Pf/Pv/Pan; Beright,”  
Done

6. P3 L124: “Central African Republic” to “CAR”; also change others
Done

Reviewer 2 Report

This is a follow-up study of REF 6 on asymptomatic carriers of malaria parasites in Central African Republic. Introduction is adequate. The Methods section needs more details and consistency. The authors should demonstrate a strong association between COVID-19 pandemic and an increase in the prevalence of asymptomatic carriage of malaria parasites if their article title is to be maintained as it is.

Major comments:

Study design:

-The authors recruited patients at a health care centre. Even if the authors included patients without signs and symptoms of malaria, these patients do not represent the general population supposedly in good health. Studies on asymptomatic carriers of malaria parasites are best conducted outside health centres and hospitals.

-Laboratory diagnosis based exclusively on rapid diagnostic test is not 100% reliable, especially in a high transmission area. Residual circulating HRP2 antigen can be present in a blood sample for several weeks after a malarial attack. Diagnosis should be confirmed by either microscopy or PCR to demonstrate the actual presence of the parasites at the time of blood collection.

-This study does not present any data on parasite density. Low parasitaemia is one of the salient features of asymptomatic malaria. Blood smears should have been prepared on-site and read by an experienced microscopist to quantify the parasites.

-The study was conducted by Polish investigators presumably with access to PCR facilities in their homeland. Laboratory diagnosis should have been confirmed by PCR, especially for P. vivax, which is supposedly uncommon in Central Africa.  

-The method for blood sample collection is not clear. In line 86, the authors mention “venipuncture procedure.” In line 108, “finger prick.”

Methods:

-Line 98-99, line 156-157, “detecting four species of Plasmodium” “mRDT tests capable of identifying four different species of Plasmodium were used in the study”: Contrary to what the authors claim, the RDT used in this study does not seem to be specific for P. ovale and P. malariae. If only the “Pan” band is positive, this RDT cannot distinguish between P. ovale and P. malariae. Please give more information on this RDT, especially its detection system (HRP2? plasmodial LDH? aldolase?). In line 155, the authors present the results as “P. ovale/P. malariae (4.8%).” This seems to support my comment concerning the inability of the RDT to make a distinction between the two species.

-Line 157, “mixed Plasmodium infections”: The authors should provide the definition (in the Methods section) of what “mixed infections” mean based on RDT result.

Data analysis:

-Line 135, Table 2: Body weight is not an independent factor with regards to age. It is in fact closely associated with age in children. A test for co-linearity should show the close relationship between these two factors in children. For this reason, age and body weight should not be analysed together by multivariate regression. The fact that the authors found a non-significant association with age and body weight in multivariate analysis (P-values, 0.634 vs 0.327) is not only expected, but it is also due to an inappropriate use of multivariate regression analysis. This point should be checked by the authors.

-The authors should also check that regression analysis is performed for a subset of data in which the study population is stratified into different age groups and sex (see the WHO criteria for anaemia).

Data interpretation:

-Lines 185-187, lines 245-246 (“prevalence of asymptomatic malaria in children...even higher than during the pre-pandemic period”): The authors observed an increase in the prevalence of asymptomatic carriers between their two studies conducted in the same population at two different time periods. The authors seem to attribute this increase in prevalence to COVID-19. The authors’ earlier study in the same study site was conducted during the month of March (2020). In that study, 35.2% of children were RDT-positive. In the present study, conducted in August-September 2021, 51.2% of children were RDT-positive. The results of the two studies are not comparable. Even in high transmission areas, some seasonal patterns are observed. A comparison of the prevalence in March to that of August-September to draw the conclusion that their results “demonstrate an increase of malaria prevalence rates by 16% over the period of 18 months (ongoing COVID-19 pandemic)” (see lines 187-188) is not scientifically sound.

Minor comments:

Line 39-40: The first citation of World Malaria Report 2021 is in line 39. REF 3 should be cited here (as REF 1).

Line 53: P. falciparum

Line 57, also line 228: eliminate malaria from the continent (please check the difference between “eradicate” and “eliminate”)

Line 64: REF 10 is cited here. What about REF 5 to REF 9? Please check reference numbering.

Line 46: coronavirus disease 2019 (COVID-19)

Line 69: Rapid diagnostic test (RDT)

Line 82: malaria rapid diagnostic test (mRDT)

Line 93: RDT (instead of “rapid diagnostic test”)

Line 94 and elsewhere: The authors should decide whether to use British or American spelling, but not both, in their manuscript (anaemia, haemoglobin).

Line 95, “haemoglobin level was measured using a portable analyzer”: This sentence can be deleted.  A similar sentence and more details are presented later (lines 106-111).

Lines 109-111: The WHO criteria for iron-deficient anaemia are much more complicated than what the authors state. The level of anaemia depends on age and sex (also whether a woman is pregnant or not). The WHO document should be cited here.

Line 191: identified

Lines 198-201, “According to Nadmukong-Nyanga et al. [21]...areas [22]”: It is not clear which reference(s) is/are supporting this statement.

Lines 203-204, “The mean body temperature in asymptomatic children with RDT(+)...the same as in children with RDT(-)”: This sentence should be deleted. The authors included patients without fever.

Line 210: What is “common anaemia”?

Line 211: Nzobo et al.

Lines 236-240: We have much to learn from COVID-19. The reasons why sub-Saharan Africa has been relatively spared from the virus are still not well elucidated. There may also be a still unknown genetic factor. The authors should probably temper their statements.

Line 250: in malaria-related

References should be in the same format throughout. For article titles, either with the first letters of each word in capital letter or only the first letter of the first word and proper nouns should be in capital letters.

REF 13: Senegalese children; Malar J.

REF 20: Plasmodium falciparum (in italics)

Author Response

Dear Reviewer,

Thank you very much for your very professional review and valuable comments.
We present the corrections made below:

Comments and Suggestions for Authors
This is a follow-up study of REF 6 on asymptomatic carriers of malaria parasites in Central African Republic. Introduction is adequate.
The Methods section needs more details and consistency. The authors should demonstrate a strong association between COVID-19 pandemic and an increase in the prevalence of asymptomatic carriage of malaria parasites if their article title is to be maintained as it is.
Correction: conclusions in Abstract (Line 25-27):
The increase in the spread of malaria in Africa is, in our opinion, associated with a well-known reduction of diagnosis, treatment and prevention of malaria. The COVID-19 pandemic is a secondary issue with the impact of reducing aid to Africa from developed countries. So, the increase in the prevalence of malaria not ‘due to’ pandemic, but ‘in the time’ of pandemic.

Study design:
- The authors recruited patients at a health care centre. Even if the authors included patients without signs and symptoms of malaria, these patients do not represent the general population supposedly in good health. Studies on asymptomatic carriers of malaria parasites are best conducted outside health centres and hospitals.
Revision: Line 83-87: Adults and children were recruited in the villages, then directed to the local health care facility in Monasao, where mRDTs were used (the only available diagnostic tool for malaria detection, no possibility of malaria confirmation by microsopy).

-Laboratory diagnosis based exclusively on rapid diagnostic test is not 100% reliable, especially in a high transmission area. Residual circulating HRP2 antigen can be present in a blood sample for several weeks after a malarial attack. Diagnosis should be confirmed by either microscopy or PCR to demonstrate the actual presence of the parasites at the time of blood collection.
Authors’ comment: RDTs were the only diagnostic tool in the CAR forest ecosystem, 180 km to the nearest medical center with a microscope, 750 km to a medical center with PCR.
The reviewer is absolutely right that HRP2 antigen can be present in a blood sample for several weeks after a malarial attack. The aim of the study was to assess the prevalence of asymptomatic malaria. The exlusion criteria was the presence of malaria signs and symptoms in the past 28 days.

- This study does not present any data on parasite density. Low parasitaemia is one of the salient features of asymptomatic malaria. Blood smears should have been prepared on-site and read by an experienced microscopist to quantify the parasites.
Authors’ comment: RDTs were the only diagnostic tool in the study, with no possibility of malaria confirmation by microscopy.

- The study was conducted by Polish investigators presumably with access to PCR facilities in their homeland. Laboratory diagnosis should have been confirmed by PCR, especially for P. vivax, which is supposedly uncommon in Central Africa.  
Authors’ comment:
The present study had a low budget without the possibility of extending the diagnostics to molecular biology (PCR).
The previous study performed among symptomatic malaria patients in 2018 (n=540) in the Central African Republic (E. Bylicka-Szczepanowska, K. Korzeniewski, A. Lass. Prevalence of Plasmodium spp. in symptomatic BaAka Pygmies inhabiting the rural Dzanga Sangha region of the Central Africa Republic. Ann. Agric. Environ. Med. 2021, 28(3), 483-90) detected 40.5% of P. falciparum infections in patients tested with RDT (CareStart Malaria Pf/HRP2/Ag). Molecular tests (PCR) confirmed P. falciparum in 94.8% of the samples, and also P. malariae (11.1%), P. ovale (9.8%), P. vivax (0.7%).

- The method for blood sample collection is not clear. In line 86, the authors mention “venipuncture procedure.” In line 108, “finger prick.”  
Correction: examination of vein blood samples

Methods:
- Line 98-99, line 156-157, “detecting four species of Plasmodium” “mRDT tests capable of identifying four different species of Plasmodium were used in the study”: Contrary to what the authors claim, the RDT used in this study does not seem to be specific for P. ovale and P. malariae. If only the “Pan” band is positive, this RDT cannot distinguish between P. ovale and P. malariae. Please give more information on this RDT, especially its detection system (HRP2? plasmodial LDH? aldolase?). In line 155, the authors present the results as “P. ovale/P. malariae (4.8%).” This seems to support my comment concerning the inability of the RDT to make a distinction between the two species.
Authors’ comment: Yes, it is not possible to distinguish between P. malariae and P. ovale using mRDT (Pf/Pv/Pan; Beright, Hangzhou AllTest Biotech Co., Ltd.).
Correction in lines: 19, 20, 102, 103, 161-163

- Line 157, “mixed Plasmodium infections”: The authors should provide the definition (in the Methods section) of what “mixed infections” mean based on RDT result.
Authors’ comment: Mixed Plasmodium infection is the existence of more than one Plasmodium species in single human at the same time. Mixed infections involving more than one species of Plasmodium may occur in areas of high endemicity and multiple circulating malarial species.
Correction in 163/164 line: More than one Plasmodium species were observed in 3.6% of the children and 11.6% of the adults tested.

Data analysis:
- Line 135, Table 2: Body weight is not an independent factor with regards to age. It is in fact closely associated with age in children. A test for co-linearity should show the close relationship between these two factors in children. For this reason, age and body weight should not be analysed together by multivariate regression. The fact that the authors found a non-significant association with age and body weight in multivariate analysis (P-values, 0.634 vs 0.327) is not only expected, but it is also due to an inappropriate use of multivariate regression analysis. This point should be checked by the authors.
- The authors should also check that regression analysis is performed for a subset of data in which the study population is stratified into different age groups and sex (see the WHO criteria for anaemia).
Correction:
Regression analysis (Table 2, 4) has been removed from the results.

Data interpretation:
- Lines 185-187, lines 245-246 (“prevalence of asymptomatic malaria in children...even higher than during the pre-pandemic period”): The authors observed an increase in the prevalence of asymptomatic carriers between their two studies conducted in the same population at two different time periods. The authors seem to attribute this increase in prevalence to COVID-19. The authors’ earlier study in the same study site was conducted during the month of March (2020). In that study, 35.2% of children were RDT-positive. In the present study, conducted in August-September 2021, 51.2% of children were RDT-positive. The results of the two studies are not comparable. Even in high transmission areas, some seasonal patterns are observed. A comparison of the prevalence in March to that of August-September to draw the conclusion that their results “demonstrate an increase of malaria prevalence rates by 16% over the period of 18 months (ongoing COVID-19 pandemic)” (see lines 187-188) is not scientifically sound. 
Correction: Line 193-195: The results demonstrate very high malaria prevalence rates in the child population living in Central Africa.

Minor comments:
-
Line 39-40: The first citation of World Malaria Report 2021 is in line 39. REF 3 should be cited here (as REF 1).  Done

- Line 53: P. falciparum  Done

- Line 57, also line 228: eliminate malaria from the continent (please check the difference between “eradicate” and “eliminate”)  Corrected

- Line 64: REF 10 is cited here. What about REF 5 to REF 9? Please check reference numbering.  Corrected

- Line 46: coronavirus disease 2019 (COVID-19)  Done

- Line 69: Rapid diagnostic test (RDT)  Done

- Line 82: malaria rapid diagnostic test (mRDT)  Done

- Line 93: RDT (instead of “rapid diagnostic test”)  Done

- Line 94 and elsewhere: The authors should decide whether to use British or American spelling, but not both, in their manuscript (anaemia, haemoglobin).  Corrected

- Line 95, “haemoglobin level was measured using a portable analyzer”: This sentence can be deleted.  A similar sentence and more details are presented later (lines 106-111).  Corrected

- Lines 109-111: The WHO criteria for iron-deficient anaemia are much more complicated than what the authors state. The level of anaemia depends on age and sex (also whether a woman is pregnant or not). The WHO document should be cited here.  Done

- Line 191: identified  Done

- Lines 198-201, “According to Nadmukong-Nyanga et al. [21]...areas [22]”: It is not clear which reference(s) is/are supporting this statement.  Corrected

- Lines 203-204, “The mean body temperature in asymptomatic children with RDT(+)...the same as in children with RDT(-)”: This sentence should be deleted. The authors included patients without fever.  Done

- Line 210: What is “common anaemia”?  (frequent) Corrected

- Line 211: Nzobo et al.  Corrected

- Lines 236-240: We have much to learn from COVID-19. The reasons why sub-Saharan Africa has been relatively spared from the virus are still not well elucidated. There may also be a still unknown genetic factor. The authors should probably temper their statements.  Corrected

- Line 250: in malaria-related  Corrected

- References should be in the same format throughout. For article titles, either with the first letters of each word in capital letter or only the first letter of the first word and proper nouns should be in capital letters.  Corrected

- REF 13: Senegalese children; Malar J.  Corrected

- REF 20: Plasmodium falciparum (in italics)  Corrected

Reviewer 3 Report

Authors used a descriptive study to identify the prevalence of asymptomatic malaria infections in Central Africa Republic. Because burden of malaria is concentrated in African countries, it may be meaningful to show the current situation of prevalence in the COVID-19 pandemic. However, this article has not been fully answered some of questions due to the insufficient description.

First, authors suggested “an increase of malaria prevalence rates by 16% over the period of 18 months” (P6L188) as the main finding, but they did not show a statistical analysis of the comparison. As mentioned by authors, the age may be associated with the prevalence, suggesting that sampling may affect the prevalence of asymptomatic malaria infections (i.e. sampling bias). Authors should conduct a statistical analysis between a previous study and current study adjusting age and other potential confounding factors.

Second, authors may suggest that they defined asymptomatic malaria infection as the prevalence of only 2 species (i.e. P. falciparum/P. vivax/Pan) (P2L99) without justification. As the definition may directly affect the prevalence, authors should justify why authors defined it that way.

Third, authors conducted multivariable regression analyses in table 2 and table 4, but there is no description in the method section and the result section. Without description of method, it may be difficult for readers to understand what authors did. Authors should add description for method of statistical analysis in the method section as well as the description of the results in the result section.

Fourth, authors showed the comparison between children and adults as follows “The prevalence of Plasmodium infections was much lower among asymptomatic 150 adults (n = 353) than it was among children” (P5L150), but they did not conduct statistical analysis for it. Authors should add the result of statistical analysis for the comparison in the result section.

Fifth, authors showed the prevalence by the four species in the results section without tables (P5L154). The prevalence may be affected by age and other potential confounding factors, it may be meaningful to show an association between the prevalence by species and these confounding factors in tables. Authors should add the statistical analysis of the association as tables.

Finally, authors did not cite a reference for some of descriptions as follows; “According to WHO, Plasmodium falciparum is responsible for 99.7% of all malaria infections in Africa. In 2020 there were a total of 215 million suspected, presumed 54 or confirmed cases of malaria on the continent (P2L53)”, “some experts estimate that a lot of malaria cases in Africa go undetected (this is associated with a high proportion of asymptomatic infections) (P2L55)”, “the prevalence rates of SARS-CoV-2 infections are much lower in African countries than they are in Asia, Europe or in North and South America (P5L164)”, “tourism industry experts forecast that Africa may soon become the leading destination for travellers from high-income countries (P6L166)”, “The WHO vaccination strategy assumed that a minimum of 40% of each country’s population would receive full primary immunization against SARS-CoV-2 by the end of 2021 (P7L229)”, “According to WHO, Africa is the least affected region globally in terms of the number of reported SARS-CoV-2 infections (1.5%) and the number of COVID-19 deaths (0.1%). (P7L232)”, “Africa’s low morbidity and mortality rates are primarily due to a lower population median age and a lower obesity prevalence compared to high-income countries in Europe or North America. (P7L236)” and “They can also be associated with different climate conditions (hot climate). (P7L238)”. Authors should cite references for these descriptions.

Minor comments

  1. “CI” in P3L115 should be spelled out.
  2. “inmalaria” (P7L250) may be typo.

Author Response

Dear Reviewer,

Thank you very much for your very professional review and valuable comments.

We present the corrections made below:

- First, authors suggested “an increase of malaria prevalence rates by 16% over the period of 18 months” (P6L188) as the main finding, but they did not show a statistical analysis of the comparison. As mentioned by authors, the age may be associated with the prevalence, suggesting that sampling may affect the prevalence of asymptomatic malaria infections (i.e. sampling bias). Authors should conduct a statistical analysis between a previous study and current study adjusting age and other potential confounding factors.

The results of this study conducted during the COVID-19 pandemic (August–September 2021) revealed that as much as 51.2% of the involved pediatric group (i.e. 162 seemingly healthy  children aged 1–15 years old living in the south–west Central African Republic) had asymptomatic malaria. A study conducted before the pandemic (early 2020) in the same group of children (500 children aged 1–15) identified asymptomatic cases of malaria in 35.2% of the examined group. The results demonstrate very high malaria prevalence rates in the child population living in Central Africa.
Comments/Correction Line 25-28: The authors abandoned the thesis about the impact of the COVID-19 pandemic (as a disease) on the increase in malaria prevalence in Africa, suggesting an indirect impact such as limited access to diagnostics, treatment and prevention of malaria due to the global pandemic and less aid from developed countries to Africa (the manuscript title: asymptomatic malaria infections in the time of COVID-19 pandemic).

- Second, authors may suggest that they defined asymptomatic malaria infection as the prevalence of only 2 species (i.e. P. falciparum/P. vivax/Pan) (P2L99) without justification. As the definition may directly affect the prevalence, authors should justify why authors defined it that way.

Correction Line 103-106: Asymptomatic malaria was defined as the presence of Plasmodium species on mRDT with no clinical signs of the disease (body temperature ≤37.5oC, no chills, headache, joint pain, weakness, vomiting or diarrhea).

- Third, authors conducted multivariable regression analyses in table 2 and table 4, but there is no description in the method section and the result section. Without description of method, it may be difficult for readers to understand what authors did. Authors should add description for method of statistical analysis in the method section as well as the description of the results in the result section.

Correction Line 145, 153: Regression analysis (Table 2, 4) has been removed from the results (applied regression did not provide additional information from the point of view of statistical analysis).

- Fourth, authors showed the comparison between children and adults as follows “The prevalence of Plasmodium infections was much lower among asymptomatic adults (n = 353) than it was among children” (P5L150), but they did not conduct statistical analysis for it. Authors should add the result of statistical analysis for the comparison in the result section.

Corrected: Statistical Methods (Line 121) and Results (Line 161-163)

- Fifth, authors showed the prevalence by the four species in the results section without tables (P5L154). The prevalence may be affected by age and other potential confounding factors, it may be meaningful to show an association between the prevalence by species and these confounding factors in tables. Authors should add the statistical analysis of the association as tables.

New Table 3 – Children with mRDT (+) (n = 83) (Line 156)
and new Table 4 – Adults with mRDT (+) (n = 43) (Line 158)

- Finally, authors did not cite a reference for some of descriptions as follows: “According to WHO, P. falciparum is responsible for 99.7% of all malaria infections in Africa (P2L55). Done

In 2020 there were a total of 215 million suspected, presumed 54 or confirmed cases of malaria on the continent (P2L56). Done

“some experts estimate that a lot of malaria cases in Africa go undetected (this is associated with a high proportion of asymptomatic infections) (P2L59)” Done

“the prevalence rates of SARS-CoV-2 infections are much lower in African countries than they are in Asia, Europe or in North and South America (P5L172)”  Done

“tourism industry experts forecast that Africa may soon become the leading destination for travellers from high-income countries (P6L174)” Done

“The WHO vaccination strategy assumed that a minimum of 40% of each country’s population would receive full primary immunization against SARS-CoV-2 by the end of 2021 (P7L239)” Done

“According to WHO, Africa is the least affected region globally in terms of the number of reported SARS-CoV-2 infections (1.5%) and the number of COVID-19 deaths (0.1%). (P7L242)” Done

“Africa’s low morbidity and mortality rates are primarily due to a lower population median age and a lower obesity prevalence compared to high-income countries in Europe or North America. (P7L246)” Done

“They can also be associated with different climate conditions (hot climate). (P7L246-247)”. Removed

Authors should cite references for these descriptions. Done

Minor comments
“CI” in P3L115 should be spelled out.  Done
“inmalaria” (P7L250) may be typo.  Corrected

Round 2

Reviewer 2 Report

Second round review

The authors took into consideration all my major and minor comments in my first round review. The revised paper is much clearer now. 

Major comments:

It is regrettable that the authors did not perform microscopy and PCR to confirm malaria diagnosis. A paragraph on the limitations of the study should be added at the end of Discussion: lack of microscopy data and PCR confirmation of diagnosis, long-term positivity of HRP2-based RDT even after complete elimination of P. falciparum from a patient with effective treatment, relatively low sensitivity of RDT compared to PCR which may lead to false negative result and an underestimation of the prevalence rate of asymptomatic Pf carriers, etc.

Minor comments:

Line 35: it is possible to identify…

Lines 42-43 and 61-62: The authors mention five African countries where most malaria infections and malaria-associated deaths occur: “> 50% of global malaria infections are reported from only five countries: Nigeria,…” and “with Nigeria… accounting for > 50% of deaths.” Do the authors mean >50% of malaria infections in lines 42-43 and >50% of death in lines 61-62 occur in five African countries? Please double check for accuracy based on WHO documents.

Line 133: venous blood sample

Line 287: A period after “milder signs and symptoms.” Then, start a new sentence “They may also be asymptomatic…”

Author Response

Dear Reviewer,
Thank you very much for your review and valuable comments.

Comments and Suggestions for Authors:
The authors took into consideration all my major and minor comments in my first round review. The revised paper is much clearer now. 

Major comments: It is regrettable that the authors did not perform microscopy and PCR to confirm malaria diagnosis. A paragraph on the limitations of the study should be added at the end of Discussion: lack of microscopy data and PCR confirmation of diagnosis, long-term positivity of HRP2-based RDT even after complete elimination of P. falciparum from a patient with effective treatment, relatively low sensitivity of RDT compared to PCR which may lead to false negative result and an underestimation of the prevalence rate of asymptomatic Pf carriers, etc.

Limitations of the study
Lines 246-251: no information on cases of malaria among pregnant women in the study group, lack of microscopy data and PCR confirmation of malaria infections (low sensitivity of RDT compared to PCR may lead to false negative results and an underestimation of the prevalence rate of asymptomatic Plasmodium carriers), possible long-term positivity of HRP2-based mRDTs even after complete elimination of P. falciparum from patients with effective treatment.  

Minor comments:
Line 35: it is possible to identify… Corrected

Lines 42-43 and 61-62: The authors mention five African countries where most malaria infections and malaria-associated deaths occur: “> 50% of global malaria infections are reported from only five countries: Nigeria,…” and “with Nigeria… accounting for > 50% of deaths.” Do the authors mean >50% of malaria infections in lines 42-43 and >50% of death in lines 61-62 occur in five African countries? Please double check for accuracy based on WHO documents.

Authors’ comment:
Yes, based on WHO World Malaria Report 2021, in the five African countries mentioned in the article (Nigeria, the Democratic Republic of the Congo, Uganda, Mozambique, and Angola) >50% world malaria infections and >50% world malaria deaths are reported.In detail:
malaria infections (Nigeria 27%, the Democratic Republic of the Congo 12%, Uganda 5%, Mozambique 4%, Angola 3.4%)
malaria deaths (Nigeria 27%, the Democratic Republic of the Congo 12%, Uganda 5%, Mozambique 4%, Angola 3%)

Line 133: venous blood sample  Corrected (Line 109)

Line 287: A period after “milder signs and symptoms.” Then, start a new sentence “They may also be asymptomatic…” Corrected (Line 225)

Reviewer 3 Report

Authors revised the manuscript, but I have only a minor comment.

Minor comments

  1. “Pf”, “Pv” and “PAN” in table 3 and table 4 should be spelled out in the tables.

Author Response

Dear Reviewer,
Thank you very much for your review and valuable comments.

Comments and Suggestions for Authors:
Authors revised the manuscript, but I have only a minor comment.

Minor comments

“Pf”, “Pv” and “PAN” in table 3 and table 4 should be spelled out in the tables.
 Corrected